# Mechanical Properties of V-O-N Coatings Synthesized by Cathodic Arc Evaporation

**DOI:** 10.3390/ma17020419

**Published:** 2024-01-14

**Authors:** Bogdan Warcholinski, Adam Gilewicz, Alexandr S. Kuprin, Galina N. Tolmachova, Elena N. Reshetnyak, Ilya O. Klimenko, Igor V. Kolodiy, Ruslan L. Vasilenko, Maria Tarnowska

**Affiliations:** 1Faculty of Mechanical Engineering, Koszalin University of Technology, 75-453 Koszalin, Poland; adam.gilewicz@tu.koszalin.pl; 2National Science Center Kharkiv Institute of Physics and Technology, 61000 Kharkiv, Ukraine; alexkuprin1983@gmail.com (A.S.K.); nanohardness@gmail.com (G.N.T.); olena.reshetniak@gmail.com (E.N.R.); ilyaklimenko91@gmail.com (I.O.K.); kolodiy@kipt.kharkov.ua (I.V.K.); rvasilenk@ukr.net (R.L.V.); 3Centre for Biological Treatment, Independent Public Clinical Hospital No. 1, Pomeranian Medical University in Szczecin, 70-204 Szczecin, Poland; m.tarnowska@spsk1.szn.pl

**Keywords:** PVD, V-O-N coating, hardness, adhesion, wear

## Abstract

The V-O-N coating set was produced at different relative oxygen concentrations of O_2_(x) = O_2_/(N_2_ +O_2_) using cathodic arc evaporation. The aim of the research was to determine the effect of oxygen on coating properties. The coatings’ composition and structural properties (X-ray diffraction (XRD), scanning electron microscopy (SEM), energy dispersive X-ray spectroscopy (EDX)) and mechanical properties—hardness, adhesion, and wear resistance (nano-indenter, scratch tester, ball-on-disc tester)—were extensively investigated. EDX and XRD analyses indicate that in coatings formed with a relative oxygen concentration in the range of 20–30%, the oxygen concentration in the coating increases dramatically from approximately 16 at.%. to 63 at.%, and the nitrogen concentration drops from about 34 at.% up to 3 at.%. This may indicate greater activity of oxygen compared to nitrogen in forming compounds with vanadium. The occurrence of the V5O9 phase belonging to the Magnéli phases was observed. Microscopic observations indicate that the number of surface defects increases with the oxygen concentration in the coating. The opposite effect is characterized by mechanical properties—hardness, adhesion, and wear resistance decrease with increasing oxygen concentration in the coating.

## 1. Introduction

Transition metal nitrides have been used in the industry as protective coatings for many years due to their good mechanical properties—high hardness, adhesion to the substrate, resistance to wear by friction, and chemical resistance to corrosion.

A slight intentional change in the chemical composition of the coating by introducing another element can lead to a change and often an improvement in their properties [1,2,3]. The effect of doping transition metal nitrides with oxygen is interesting, as it can affect their optical and decorative properties [3]. Their strength properties are intermediate between transition metal oxides and transition metal nitrides [4]. Due to the higher reactivity of oxygen compared to nitrogen, ionic metal–oxygen bonds are formed even at low oxygen concentrations. Such oxides can be formed in the presence of a growing metal nitride layer.

Depending on the concentration of oxygen in the Me-O-N (Me—transition metal) coating, its properties, including color, may change. In the best-known compounds Zr-O-N and Ti-O-N, the color changes from light yellow to dark blue with increasing oxygen concentration [3,5]. Cr-O-N, on the other hand, does not change its color. Changing the chemical composition of the Me-O-N coating as a result of changing the ratio of partial pressures of oxygen and nitrogen during their formation leads to the formation of new structures characterized by different properties [6]. The topic of transition metal oxynitrides is much less present in the literature than the topic of transition metal nitrides. However, their potential applications as diffusion barriers, decorative coatings, temperature-dependent resistors, new pigments, etc. [7] result in publications about Me-O-N (where Me is Cr, Nb, Ti, Ta, W, Mo, Zr) showing both mechanical, including tribological, and optical properties [7,8,9,10,11,12,13].

One of the less frequently studied nitrides is vanadium nitride, which is characterized by excellent mechanical properties, including wear resistance, temperature stability, and catalytic activity [14], hence its wide applications, including protective coatings for tools [15] and implants [16]. It has also found application in microelectronics [17] as well as decorative coating.

However, a much larger field of application has been found for vanadium oxynitride. It has been used as a durable catalyst for the electrochemical reduction of nitrogen to ammonia [18,19] and pseudo-capacitive materials for electrochemical capacitors [20]. These coatings are characterized by good optical properties (transmittance over 40% in the visible range) and anti-corrosion properties [21]. However, the amount of data on the mechanical properties of V-O-N coatings is small, and there is no information on the hardness, adhesion of the coatings, and their wear resistance, parameters taken into account when applying the coatings as decorative ones. It is known that doping the VN coating with C [22] or Ag [23] causes a decrease in hardness, but the tribological properties are the opposite. This causes difficulties in predicting the mechanical properties of oxygen-doped VN coatings. The change in the structure of V-N coatings as a result of their oxidation is the most frequently studied [24]. V-O-N coatings have been formed so far by the magnetron sputtering method [21]. 

Due to their properties, thin vanadium nitride coatings are used as protective and lubricating coatings. Oxidation of vanadium nitride occurring in the friction process (especially at high temperatures) significantly affects the tribological properties [25,26].

The aim of this research is to assess the influence of oxygen concentration in the V-O-N coating on its mechanical properties. The coatings were formed by cathodic arc evaporation. To our knowledge, this method has not been used to synthesize vanadium oxynitrides so far.

## 2. Materials and Methods

### 2.1. Coating Deposition

A set of VN, V-O-N, and VO_x_ coatings was formed using unfiltered cathodic arc evaporation (CAE) in a “Bulat” (Kharkiv, Ukraine) system equipped with a V (99.99%) cathode of 60 mm diameter. A cylinder with an inner diameter of 600 mm and length of 700 mm was the vacuum chamber of the system. A vacuum-arc plasma source with magnetic stabilization of a cathode spot was used [27]. HS6-5-2 steel (PN EN ISO 4957; 2018-09 [28]) substrates, 32 mm in diameter and 3 mm thick with the chemical composition (wt.%): C—0.87, W—6.4, Mo—5.0, V—1.9, Cr—4.2, Mn—0.3, Si—0.4 and Fe (balance) were used. They were polished using abrasive silicon carbide paper to a roughness Ra of about 0.02 μm. The substrates were chemically degreased and ultrasonically cleaned in a hot alkaline bath for 10 min and dried in warm air. After cleaning, they were mounted on a planetary rotating holder with a rotation speed of about 9 rpm. The substrate–cathode distance was about 400 mm. 

In the first step of coating deposition, the chamber was evacuated to a pressure of 2 × 10^−3^ Pa. Substrates were ion etched with vanadium ion bombardment by applying a DC bias of −1300 V for 3 min. The arc current was 90 A.

The second step was improvement of the adhesion of the coatings. A pure vanadium layer (about 0.1 μm thick) was deposited on the substrate at the bias voltage of −100 V for 5 min. 

The third step was the deposition of the V-O-N coatings. They were deposited in a (N_2_ +O_2_) gas mixture with different relative oxygen concentrations O_2_(x) = O_2_/(N_2_ +O_2_)%, where x equals 0, 10, 20, 30, 50, 70 and 100%. During deposition, the (N_2_ +O_2_) gas mixture pressure and substrate temperature were kept at 1.5 Pa and about 400 °C, respectively. The deposition was performed at a substrate bias voltage of −150 V. The deposition time was 60 min in all cases.

The samples were designated V-O(x)-N, where x is the relative oxygen concentration during coating formation.

### 2.2. Coating Investigations

The coatings formed at different relative oxygen concentrations in the working chamber were characterized by the following:Thickness—Calotest;Surface morphology—scanning microscope, QUANTA 200, FEI Company, Hillsboro, OR, USA)Crystalline structure—X-ray diffractometer, DRON-4-07, (Bourevestnik, Saint Petersburg, Russia), U = 35 kV, I = 20 mA, Cu-Kα radiation λ = 0.154187 nm and a nickel selectively absorbing filter, focusing by Bragg-Brentano, step size 0.05°, counting time 10 s. PCPDFWIN data were used to identify the phase composition;The microstructure and chemical composition of the coatings—scanning electron microscope (JSM-7001F, JEOL Ltd., Tokyo, Japan)) equipped with EDS (Energy Dispersive X-ray Spectroscopy, INCA ENERGY 350, OXFORD Instruments, Abingdon, United Kingdom) (20 kV). The elements were analyzed with an accuracy of about 0.5 at.% (vanadium) and 1.0 at.% (nitrogen and oxygen);Mechanical properties, such as Young’s modulus and hardness—Nano-Indenter G200 system (Agilent Technologies, Santa Clara, CA, USA) automatic nano-hardness tester equipped with Berkovich diamond tip. The indentation depth was fixed at 0.3 µm less than 0.1 coating thickness, which enables correct coating hardness measurement. The average values are from 20 measurements;Adhesion—scratch tester (CSEM Revetest, (CSM Instruments, Peseux, Switzerland)), and the following measurement parameters were used: indenter speed 10 mm/min, distance 10 mm (5 mm), linear change in normal load from 0 to 100 N (50 or 150 N) at a speed of 50 (100 and 150) N/min. Based on microscopic observations, two critical loads were determined: Lc_1_—the first lateral cracking occurs, and Lc_2_—the complete delamination of the coating occurs. These loads were determined as the average of at least 3 measurements;Friction and wear—ball-on-disc, normal force 20 N, sliding speed 0.2 m/s, distance 200 m to 2000 m dependent on coating hardness, humidity 40%. Counterpart—Al_2_O_3_ ball with a diameter of 10 mm. The friction process was carried out three times under the same conditions. The wear track profile (to determine wear volume) was measured 4 times every 90° for each friction track;The tested samples were not subject to any special preparation. They were tested in the condition they were in after being removed from the working chamber. To obtain a coating fracture, the steel substrate was cut to a depth of approximately half its thickness. After cooling it in liquid nitrogen, the coated substrate broke.

## 3. Results

The color of the coatings depended on the relative oxygen concentration during their formation, Figure 1. The VN (without oxygen) coating is light gray as CrN, Figure 1a. Oxygen-doped coatings change color from yellow-light brown (Figure 1b) to dark brown, gradually turning to black. An increase in the oxygen flow rate during coating formation favors the replacement of nitrogen with oxygen in the vanadium nitride. This may result in a reduction in the number of free electrons in the coating material and, thus, a change in its color.

Due to the color change in the V-O-N coating due to its chemical composition, they have great decorative value. Color is probably one of the most important parameters for the decoration industry [29]. This aesthetic value can make a perceived object luxurious and desirable.

### 3.1. Deposition Rate

In Figure 2, the dependence of the deposition rate as a function of relative oxygen concentration is shown. There is a noticeable increase in the deposition rate with the amount of oxygen in the working chamber during the formation of the coating. The opposite effect was observed in ZrN_x_O_y_ coatings formed by reactive rf magnetron sputtering [13].

### 3.2. Coatings Morphology

On the surface of the coating, Figure 3, many macroparticles or craters can be seen as a result of their removal or as a shadowing effect during deposition. They are surface defects and are characteristic of the deposition method of the coating formation—cathodic arc evaporation. These defects cause an increase in the surface roughness of the coating and deterioration of its quality compared to coatings formed by magnetron sputtering [30].

Surface defects (macroparticles) have oval shapes similar to spheres of various dimensions. About 80% of the macroparticles have a diameter of not more than 1.5 µm. The remaining macroparticles are larger, but only about 10% are larger than 3.0 µm. It can also be noticed that the number of defects on the surface of the coatings increases with the increase in the relative concentration of oxygen.

Typical cross-sections of the respective coatings are also provided as inserts in Figure 3. A dense and compact and columnar structure of coatings without porosity at the grain boundary can be noticed. The structure of the deposited coating can be classified as zone II according to the Thornton model, which is characterized by columnar grains and the occurrence of surface micro-roughness. At higher substrate temperatures, the mobility of the atoms increases. More nuclei are formed, which then grow into a compact columnar structure. Evaporated drops can also be observed on the surface of the coating formed, which have been partially embedded in the coating, Figure 3. The presence of solidified drops of the evaporated target on the surface of the coating is a characteristic feature of the deposition method used and is its disadvantage.

The amount and size of macroparticles resulting in, e.g., surface roughness, is related to the technological parameters of coating formation, reactive gas pressure, arc current, substrate bias voltage, and the cathode material. The results of studies by Munz et al. [31] indicate that the amount of cathode macroparticles deposited on the steel substrate decreases with the increase in the melting point of the cathode material.

### 3.3. Chemical and Phase Composition

The chemical and phase composition of V-O-N coatings strongly depends on the technological parameters of their formation, in this case, the concentration of oxygen in the working atmosphere. The concentration of nitrogen and oxygen in the coating is controlled by the flow rate of oxygen and nitrogen to ensure a constant pressure of the mixture of the above gases in the working chamber.

V-O-N coatings formed without oxygen have a stoichiometric composition, N/V = 1, Figure 4. The coatings formed with a relative oxygen concentration above 20% were characterized by a high oxygen concentration, above 60 at.%, and the (N + O)/V ratio is almost constant and amounts to about 2.

The chemical composition does not change linearly with increasing relative oxygen concentration in the working atmosphere in the vacuum chamber during coating formation. With a relative oxygen concentration in the range of 20–30%, the oxygen concentration in the coating increases from about 16 at.% to 63 at.%, and the nitrogen concentration decreases from about 34 at.% to 3 at.%. For a relative concentration of oxygen above 30%, the chemical composition of the coating changes slightly. This may indicate a higher activity of oxygen compared to nitrogen to form compounds with vanadium, which was previously found [13].

The crystal structure of V-O-N coatings changes as the elemental composition changes. Figure 5 shows X-ray diffraction patterns of coatings formed at different values of oxygen concentration in a gas mixture with nitrogen. According to the X-ray diffraction analysis, all the studied coatings can be divided into four types according to their phase composition: the first type is presented in Figure 5a, the second in Figure 5b,c, the third in Figure 5d,e, and the fourth type in Figure 5f,g.

The phase identification was performed according to the International Centre for Diffraction Data (ICDD) pattern. The first structural group includes V-N coatings deposited in the absence of oxygen (Figure 5a). The diffraction pattern shows very intense and relatively broad peaks corresponding to h-VN vanadium nitride (ICDD 01-080-2702) with a hexagonal structure of WC type (space group number 187). The peaks are shifted to larger angles, indicating reduced lattice parameters. The size of the nitride crystallites (coherent scattering zones) calculated using the Scherrer formula is about 12 nm.

The ratio of the intensity of the h-VN diffraction lines differs significantly from the standard data for powder diffraction, according to which the highest intensity is the reflection (101). On the diffractogram, the most intense peaks are those near 39.0° and 83.8°, which correspond to reflections (100) and (200) of h-VN. Other reflections of this phase (001) and (101) are much weaker in the diffractograms, and reflections (110), (002), (111), (102) are not detected at all, which is due to the formation of a sufficiently strong axial texture with a predominant orientation of crystallites with planes (h00) parallel to the surface of the sample, i.e., the location of the crystal lattice hexagons with the c-axis in the coating plane. A similar arrangement of hexagons was previously observed in α-Ti vacuum-arc coatings [32], but in h-VN coatings, the predominant orientation of crystallites was found to be with the base planes (00l), i.e., the c-axis perpendicular to the surface [33].

The addition of oxygen to the vacuum chamber during deposition causes a sharp change in the phase composition of nitride layers. V-O-N coatings deposited at 10% and 20% of oxygen relative concentration in a gas mixture with nitrogen 12% and oxygen 16%, respectively, should be assigned to the second type of structure.

The intensity of the peaks in the diffractograms of these coatings (Figure 5b,c) is almost an order of magnitude lower than that of the V-N coating (see the scale on the ordinate axis), and the background level is twice as high. The specific halo shape of the background indicates the formation of a significant amount of X-ray amorphous phase. The peaks in the diffractogram were identified as the c-VN_0.52_O_0.26_ phase (ICDD 37-1178), which is an ordered solid solution of oxygen based on cubic c-VN. The possibility of replacing nitrogen with oxygen (and vice versa) in the V-O-N system is due to the proximity of their atomic radii (r_N_/r_O_ ≈ 1.05) and the similarity of the structure of the corresponding phases. Vanadium oxides and nitrides are based on VN_6_ or VO_6_ octahedra with common angles and edges [34]. The change from h-VN → c-(V,O)N agrees with theoretical first-principles calculations, according to which the addition of oxygen stabilizes the cubic phase [35]. The diffractograms do not show any peaks of this phase in the range of angles 2θ 36°–45°, in particular, the most intense (according to the table data) main reflections (400) and (222). The most intense in these experiments is the superstructure reflection (332), which can be associated both with the texture in the coatings and with the peculiarities of the ordering of the solid solution. For example, TiN coatings with a cubic structure, such as NaCl deposited from high-energy vacuum-arc plasma streams, are characterized by a predominant (220) orientation, which leads to the disappearance of (h00) and (hhh) reflections in the diffractograms [36,37,38]. The peaks detected in the diffractograms of V-O-N coatings are narrower than for V-N and correspond to a crystallite size of about 35 nm. Further increase of the oxygen concentration in the gas mixture during deposition up to 30% leads to an increase in the amorphization of V-O-N coatings, which is associated with a jump in the oxygen content and a transition from a predominantly nitride structure to an oxide structure.

V-O-N films deposited at relative 30% and 50% oxygen concentration in a gas mixture with nitrogen, containing about 60% oxygen, should be assigned to the third structural type. The diffractograms of these coatings show an intense background and only a few weak peaks, the most intense of which is located near 52° (Figure 5d,e), indicating the formation of an X-ray amorphous structure with a small amount of crystalline phase species that is difficult to identify.

The formation of amorphous phases has already been observed in ion-plasma coatings of the V-O-N system. In [39], thin amorphous films of vanadium oxynitride were deposited by reactive magnetron sputtering. To obtain the crystalline phases of the oxynitrides, rapid thermal annealing in an Ar atmosphere at 600 °C and 700 °C for 5 min after deposition was necessary.

The maximum number of V_5_O_9_ lines is detected in the V-O coating diffractogram. The size of the crystallites in the oxide is about 40 nm. In several studies, X-ray diffraction shows the amorphous nature of vanadium oxide films deposited by magnetron sputtering [40,41,42] and vacuum-arc deposition [43,44], which is due to the low temperature of the substrate and the energy of the particles forming the coating.

### 3.4. Nanoindentation

The evolution of hardness and Young’s modulus depending on the relative oxygen concentration during coating formation is shown in Figure 6. The highest value (H = 40.1 ± 1.6 GPa and E = 504 ± 25 GPa) was characteristic of the coating formed without oxygen. An increase in relative oxygen concentration causes the hardness and Young’s modulus to decrease to the lowest values of hardness (0.6 ± 0.1 GPa) and Young’s modulus (32 ± 3 GPa) for a coating formed at 100% relative oxygen concentration, i.e., without nitrogen.

In recent years, the significant role of the elastic properties of the coating has been emphasized, and the ratio of hardness and modulus of elasticity: H/E [45] and H^3^/E^2^ [46] is presented as a parameter by which can predict the wear. The H/E ratio is related to the elastic deformation to failure of the coating (the coating’s susceptibility to loading), while H^3^/E^2^ is an indicator of resistance to plastic deformation, and an increase in this ratio leads to an improvement in load capacity. Plastic deformation of the coating/substrate system significantly intensifies the wear in the sliding contact. Both H/E and H^3^/E^2^ ratios show that the nature of hardness changes and decreases with oxygen concentration in the coating, Figure 6b.

### 3.5. Adhesion

Adhesion is one of the key parameters of protective coatings that determine their use. One of the most effective methods for determining adhesion, but also penetration depth or the coefficient of friction between the indenter and the substrate, is the scratch test. As a result of the increasing load on the indenter moving at a constant speed on the surface of the coating, elastic and then plastic deformations are observed. Its cracks are observed in hard coatings.

Among many parameters recorded during the scratch test, in addition to normal force, friction force, and acoustic emission, there is also the penetration depth. This measurement makes it possible to assess the resistance to coating destruction. It can be seen that coatings formed at a relative oxygen concentration of 0% or 10% have a lower penetration depth and a smoother depth profile. After exceeding the critical force Lc_1_, they have a rougher profile, which results from the occurrence of cracks in the coating, as well as the possible displacement of a certain volume of coating material in front of the indenter.

The penetration depth of the V-O(0)-N coating increases almost linearly to about 25 μm, with the normal load increasing to about 95 N, Figure 7. This is possible due to the high hardness of the coating and the plastic deformation of the steel substrate. In the V-O(20)-N coating, the linear increase in the penetration depth occurs up to about 15 µm with a normal load of about 50 N. A significant difference during the curve is observed. In the latter, it is smooth to a penetration depth of about 5 μm (for a load of about 13 N). In the other curves presented in Figure 7, the linear nature of changes in the penetration depth is much shorter, and with a normal load of about 4 N, the penetration depth increases significantly. At higher depths, the curves are wavier due to the vertical movement of the indenter during coating fracture and the horizontal movement of the coating-substrate system. An increase in the penetration depth is observed with a decrease in the hardness of the coating.

In mechanical applications, materials with high hardness are mainly used. In this case, the coating in front of the indenter is under compressive stress. The opposite state of stress occurs behind the indenter, where the coatings are stretched due to the force of friction. These stresses increase with increasing load. Therefore, when the tensile strength of the coating is exceeded, it cracks. These cracks are usually perpendicular to the surface of the coating and may propagate further at the interface between the coating and the substrate. This may lead to the formation of adhesive cracks, which are accompanied by local loosening of the coating in various areas—outside and/or inside the scratch.

Enlarging the image of the characteristic areas of the scratch indicates significant differences between the coatings. The coating formed without oxygen with a relative oxygen concentration of 0% shows excellent resistance to damage and higher critical loads (Figure 8).

In the image of the crack obtained at a critical load of about 84 N, conformal cracks of the coating are visible, indicating cohesive failure caused by substrate deformation and/or the presence of compressive residual stress inside the coating material and in front of the moving indenter [47]. In the coating formed at a relative oxygen concentration of 20%, the first damage to the coating (spallation) is observed at a normal load of about 13 N (Figure 9), while delamination at a load of about 54 N. The profile of the coating seems to be smooth; however, numerous chippings of the coating are observed at the boundary of the impact of the indenter. The crack test was carried out for this coating in the normal load range from 0 N to 100 N.

For a coating formed at a relative oxygen concentration of 50%, a small critical force of about 8 N is observed, Figure 10. The test was carried out in the normal load range from 0 N to 50 N. The first damage to the coating, large chips, occurs already at about 3 N, while delamination occurs at about 8 N.

Due to the significant difference in the value of the critical force Lc_2_ of the coatings formed at different relative oxygen concentrations, above 100 N, but also definitely below this value (Figure 11a), the scratch test was carried out in different ranges of normal load and loading rate.

The list of critical loads for all tested coatings is shown in Figure 11b. The coating formed in an oxygen-free atmosphere is characterized by a very good critical force Lc_2_ = (121 ± 11) N and decreases rapidly with increasing oxygen concentration in the coating. For coatings formed with a relative oxygen concentration of not less than 30%, the critical load remains constant, from about 8 N to about 15 N. A strong correlation with hardness can be noticed here (Figure 6a), where the above coatings are characterized by a hardness of up to 6 GPa, i.e., lower than the hardness of the substrate. A similar correlation can be seen with the results of the chemical composition of the coatings (Figure 4). Coatings for industrial applications should be characterized by good adhesion. The crack initiation of the coating in the scratch test defined by the critical load Lc_1_ indicates its resistance to cracking; hence, its high value would be advantageous. It has been shown that both high Lc_1_ and Lc_2_ are required to obtain high coating toughness [48]. Using these data, a new coating toughness parameter was proposed, called Scratch Crack Propagation Resistance (CPR_S_), CPR_s_ = Lc_1_(Lc_2_ − Lc_1_) [48]. The V-O-N coatings are characterized by different values of this parameter, Figure 11b. The highest toughness was characterized by the VN coating (over 3300 N^2^), and it decreased with oxygen concentration. The lowest values were obtained for coatings formed with a relative oxygen concentration of 50% and more, up to about 20 N^2^.

The result of the critical force in the scratch test is influenced, to a greater or lesser extent, by the thickness of the coating, the hardness of the coating and the substrate, the roughness of the coating, and even the coefficient of friction between the coating and the indenter, as well as its radius and wear. The parameters of the test itself are also important: the speed of moving the indenter and the speed of increase of the force loading the indenter [49]. For this reason, the measurements were made on coatings deposited on equally prepared substrates and in the same configuration as the measuring device.

Comparing the results of the scratch test carried out in different measurement conditions or with different thicknesses is difficult and, in extreme cases (different substrates), impossible. The same indenter speed dx/dt, loading rate dL/dt, and the same type of indenter should be used for all samples. The use of the same parameters in the crack test can be problematic when the measured values of the critical loads of the coatings differ significantly from each other. Therefore, the same type of indenter was used in the tests carried out, and the indenter speed was set at 10 mm/min.

As mentioned, the adhesion tests were carried out for different loading rates dL/dt. With the increase of the indenter speed, the critical load decreases, and for the 10 mm/s used in the work, the loading rate is almost constant in the range from 10 N/min to 50 N/min [50]. This means that the adhesion of the coatings calibrated to one loading rate should be close to the value obtained at a given dL/dt. An increase in the indenter speed reduces the critical load to 30% [49,50].

### 3.6. Friction and Wear

The V-O-N coating wear rate increases with the oxygen content in the coating, Figure 12. The lowest wear rate is for the coating formed without oxygen, (1.1 ± 0.2) × 10^−7^ mm^3^/Nm. A comparable value is shown in the coating with 12% oxygen concentration, formed in an atmosphere with a relative oxygen concentration of 10%. The increase in oxygen in the working atmosphere during the formation of the coating reduces its wear resistance. It should be emphasized that with the decrease in the hardness of the coating, the sliding distance was also reduced so that the coating was not completely worn away.

## 4. Discussion

Some transition metal nitrides (TiN and ZrN) are widely used in industry as decorative coatings due to their golden color as well as high hardness and adhesion to the substrate. Therefore, they provide good protection of tools and machine parts against premature wear, increasing their service life. It seems, based on the results obtained (Figure 1, Figure 6 and Figure 12), that V(O)N can also be added to them, the color of which depends on the concentration of oxygen in the coating. The color change occurs in a small range of relative oxygen concentration, up to about 30%, but they must have good mechanical properties—high hardness and Young’s modulus and adhesion, as well as a low wear rate.

The deposition rate (Figure 2) shows the opposite effect compared with ZrN_x_O_y_ coatings. The change in the deposition rate can be explained by the occurrence of the so-called target poisoning by the formation of oxides and nitrides of the cathode material on its surface. On the surface of the cathode, there is a competition between the formation of oxide and nitride layers and their removal by sputtering. As the concentration of oxygen in the working atmosphere increases, the formation and removal of the oxide layer, rather than the nitride, becomes more important.

The melting point of Zr (2128 K) is much lower than its ZrN nitride (3233 K) and ZrO_2_ oxide (2950 K). Therefore, the cathode can be coated with zirconium oxide and zirconium nitride. Due to the lower sputtering efficiency of zirconium oxide and nitride, the deposition rate decreases, as was reported in Ref. [13]. In the case of vanadium, the situation is reversed. The melting point of vanadium is 2183 K, while the melting points of VN and V_2_O_5_ are lower—2050 K and 958 K, respectively. Also, other vanadium oxides that can form on the cathode surface have melting points lower or close to the cathode temperature: VO (2063 K), V_2_O_3_ (2243 K), and VO_2_ (2240 K).

Although the target surface can be coated with a compound of vanadium oxide and nitride due to a lower melting point than metallic vanadium, the poisoning effect is not significant. The increase in the thickness of the coating is rather related to the fact that the density of vanadium oxide V_2_O_5_ (3350 kg/m^3^) is almost two times lower than the density of vanadium nitride (6130 kg/m^3^). Assuming a constant amount of vanadium ions emitted from the cathode, the vanadium oxide layer should be about twice as thick as the vanadium nitride layer, as can be seen in Figure 2.

The coating formed without nitrogen is characterized by a hexagonal structure, and the addition of oxygen changes the structure to a cubic one, Figure 5. Previous studies have shown that the hexagonal vanadium nitride coating is characterized by higher hardness than the coating with a cubic structure [33,51].

The values of the mechanical properties of coatings formed with a relative oxygen concentration above 20% are not high. This is probably related to the presence of an amorphous V-O-N structure (V-N and V-O diffraction lines are not visible, Figure 5d,e). A further increase in the relative concentration of oxygen enables the formation of vanadium oxide V_2_O_5_ (Figure 5e–g), whose layered type of structure significantly deteriorates the mechanical properties of the coatings.

As mentioned above, the peaks are shifted to larger angles. This is quite unexpected and requires further investigation. The coatings of transition metal nitrides formed using the vacuum-arc technique are characterized by rather increased lattice parameters in the direction of the normal to the surface due to the presence of high residual compressive stresses [52,53].

The formation of texture in coatings deposited from energy ion fluxes is considered in conjunction with stresses and is associated with the possibility of minimizing the free energy of the system consisting of strain energy and surface energy [54,55]. The texture axis varies depending on the ion energy, deposition temperature, the angle of the substrate relative to the flow of deposited particles, etc. [55,56]. The change in the predominant orientation in h-VN coatings from (00l) to (h00) is most likely due to a decrease in the substrate rotation speed from 30 [33] to 9 rpm and the substrate–cathode distance of 400 mm.

It should be noted that in VN coatings with a close stoichiometric composition, the formation of the h-VN hexagonal phase is facilitated by the high energy of the vacuum-arc plasma ions deposited on the substrate when a substrate bias voltage of −150 V is applied. The findings of Aissani et al. [14] indicate that in the magnetron sputtering process, which is characterized by lower energies of the particles forming the coating, it crystallizes in the c-VN cubic structure of the NaCl type (space group number 225). Vacuum-arc coatings deposited on a substrate at a low substrate bias voltage of −50 V have the same c-VN structure. As the bias, and therefore the ion energy, increases, the phase transformation c-VN → h-VN occurs, and in the range of 150–300 V, only h-VN is detected [33]. According to the theoretical first-principles study of the phase stability of stoichiometric vanadium nitrides, it is the WC-based VN_x_ structures for x > 0.8 that appear to be the most stable, although the c-VN_x_ phase predominates in experiments over a wide range of concentrations and synthesis temperatures [35].

The literature indicates that it is difficult to accurately determine the crystalline phase in vanadium oxynitride films due to the possibility of the simultaneous presence of oxynitrides, nitrides, and various oxides [39,40]. The phase diagram of the V-O system is rich in numerous oxides because vanadium can exist in different valence states (from +2 to +5) [57]. Additional difficulties in determining the phase composition of coatings are associated with the fact that some of the peaks from different phases often overlap in X-ray spectra. The mixed-valence oxide V_5_O_9_ belongs to the homologous series of Magnéli phases, which are metastable compounds between V_2_O_3_ and VO_2_, defined by the general stoichiometric formula V_n_O_2n−1_, where n is from 3 to 9 [57]. In many studies, X-ray diffraction shows the amorphous nature of vanadium oxide films deposited by magnetron sputtering [40,41,42] and vacuum-arc deposition [43,44], which is due to the low temperature of the substrate and the energy of the particles forming the coating. Additional post-deposition annealing is used to form polycrystalline phases [41,44]. The combination of sufficiently high values of bias voltage and substrate temperature of 150 V and 400 °C, respectively, allows us to obtain crystalline vanadium oxide-based coatings directly during deposition.

The hardness of V-O-N coatings (Figure 6a) is strongly related to their structure, Figure 5. The coatings with low oxygen concentration are characterized by high hardness, an increase in oxygen concentration, and the formation of vanadium oxide phases that cause their softening. This consequently reduces the H/E and H^3^/E^2^ ratios, Figure 6b.

As hardness is one of the wear resistance criteria, it can be assumed that coatings formed with a relative oxygen concentration of not less than 30% should be characterized by poor wear resistance, Figure 12. Their hardness, below 10 GPa, is almost two times lower than that of the Al_2_O_3_ counterpart. Also, the low CPR_S_ parameter (Figure 11b) for coatings formed with a relative oxygen concentration of not less than 30% indicates the possibility of poor wear resistance of these coatings. Vanadium nitride and other transition metal nitrides are characterized by very good adhesion to the substrate. Oxygen doping significantly impairs adhesion.

## 5. Conclusions

The article presents the results of the technology for synthesizing V-O-N coatings using different relative oxygen concentrations, as well as the results of research on the structure and their mechanical properties. It was found that with an increase in the relative oxygen concentration:The color of the coatings varies from silver (VN) through light yellow (10% oxygen), yellow-gray (20% oxygen) to black;The deposition rate of the coatings increases. The deposition rate of the V-O coating is almost twice as fast as that of the V-N coating;The number of surface defects increases;In the V-N coating, the N/V ratio amounts to 1. The coatings formed with a relative oxygen concentration above 20% were characterized by a high oxygen concentration, above 60 at.%, and the (N + O)/V ratio is almost constant and amounts to about 2. This indicates a higher activity of oxygen compared to nitrogen in the formation of vanadium compounds;An increase in oxygen concentration in the coating causes the transformation of h-VN → c-V_0.52_O_0.26_ → V_5_O_9_. This last phase is one of Magnéli’s phases;Significant reduction in hardness, Young’s modulus, as well as H/E and H^3^/E^2^ rates. The highest hardness and Young’s modulus are characterized by the coating without oxygen, 40 GPa and 500 GPa, respectively, while the lowest by the coating without nitrogen, 0.6 GPa and 32 GPA, respectively. These values strongly correlate with the phase composition of the coatings;Adhesion also shows a similar trend of changes with oxygen concentration as hardness. Although the coating without oxygen is characterized by high adhesion—about 120 N—for the coating without nitrogen, it is only about 9 N;There is also a very clear change in the wear resistance of coatings. It decreases significantly with increasing oxygen concentration. This could be predicted based on the low values of the H/E (elastic deformation to failure), H^3^/E^2^ (the resistance to plastic deformation), and CPRS (Scratch Crack Propagation Resistance) indices. This is most likely related to the phase composition of the coatings.

## Figures and Tables

**Figure 1 materials-17-00419-f001:**
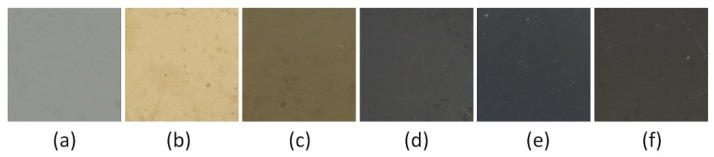
V-O-N coatings color for O_2_(x): (**a**) 0%, (**b**) 10%, (**c**) 20%, (**d**) 30%, (**e**) 50%, (**f**) 70%.

**Figure 2 materials-17-00419-f002:**
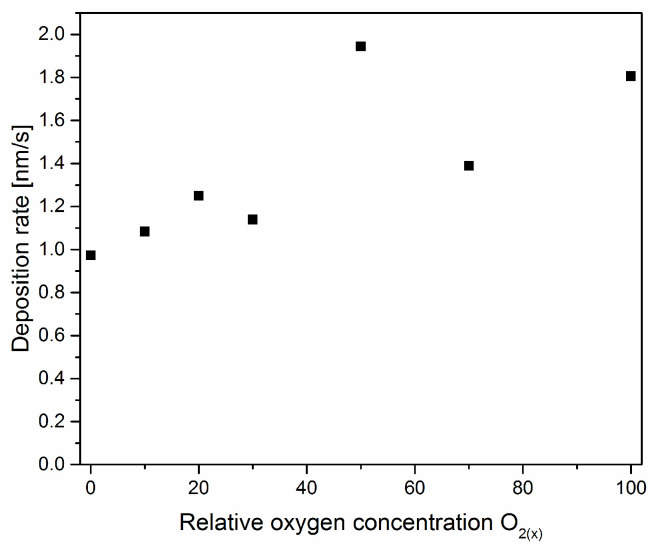
The deposition rate of V-O-N coatings formed at various relative oxygen concentrations.

**Figure 3 materials-17-00419-f003:**
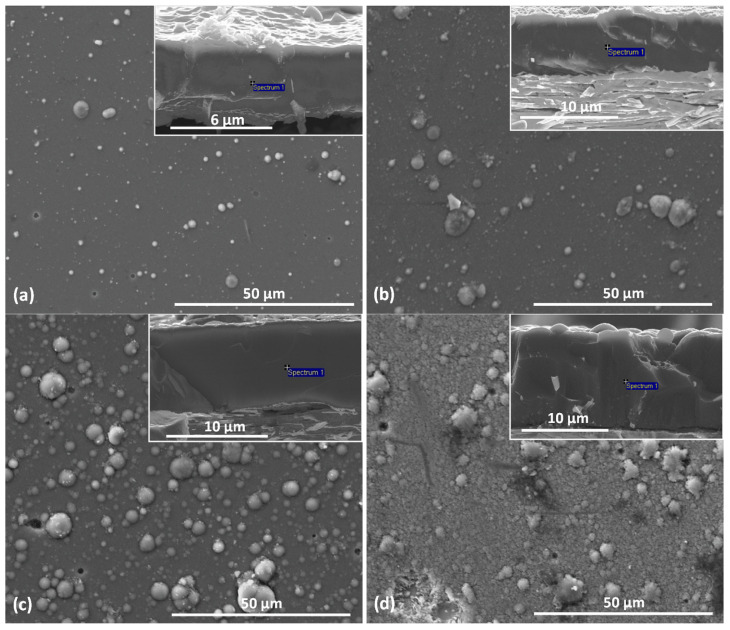
SEM pictures of surface and cross-section (inserts) of V-O-N coatings formed at relative oxygen concentration: (**a**) 10%, (**b**) 20%, (**c**) 30%, (**d**) 50%.

**Figure 4 materials-17-00419-f004:**
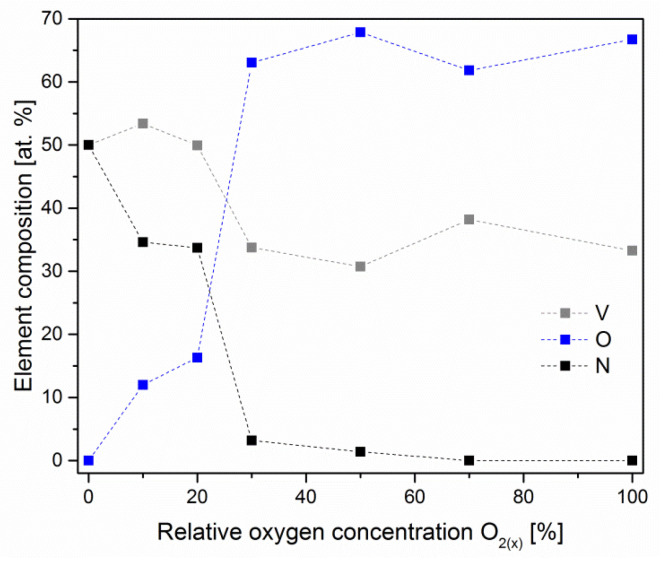
Chemical composition of V-O-N coatings formed at various relative oxygen concentrations. The dotted lines are only for eye guidance.

**Figure 5 materials-17-00419-f005:**
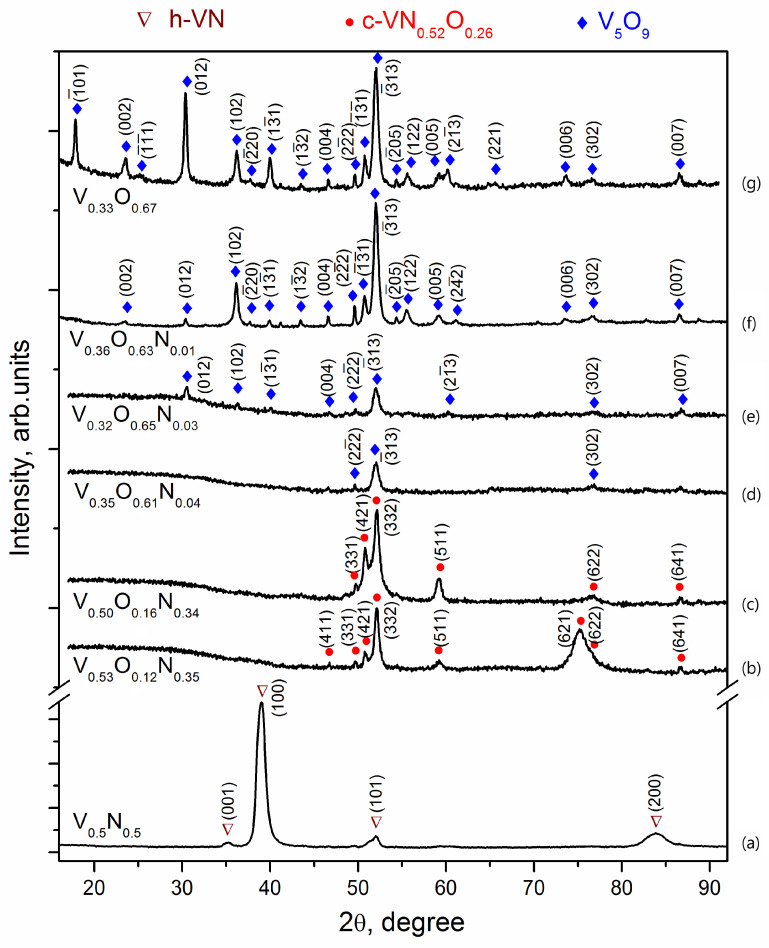
XRD diffractograms of V-O-N coatings deposited at relative oxygen concentration: 0% (**a**), 10% (**b**), 20% (**c**), 30% (**d**), 50% (**e**), 70% (**f**), 100% (**g**).

**Figure 6 materials-17-00419-f006:**
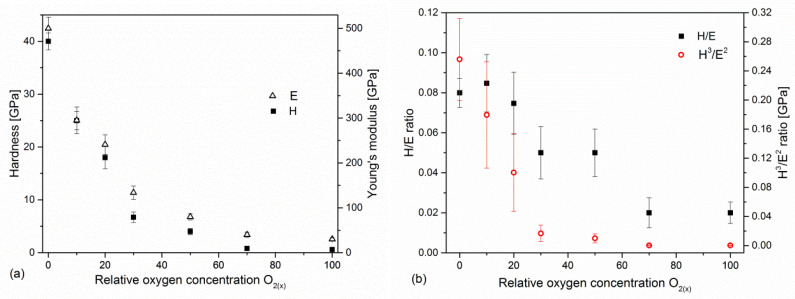
Hardness and Young’s modulus (**a**) and H/E ratios (**b**) of V-O-N coatings formed at various relative oxygen concentrations.

**Figure 7 materials-17-00419-f007:**
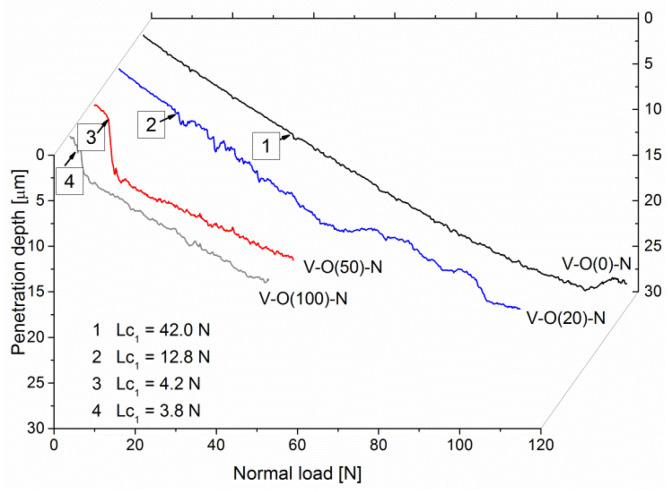
The penetration depth of selected V-O-N coatings formed at various relative oxygen concentrations dependent on normal load during scratch test.

**Figure 8 materials-17-00419-f008:**
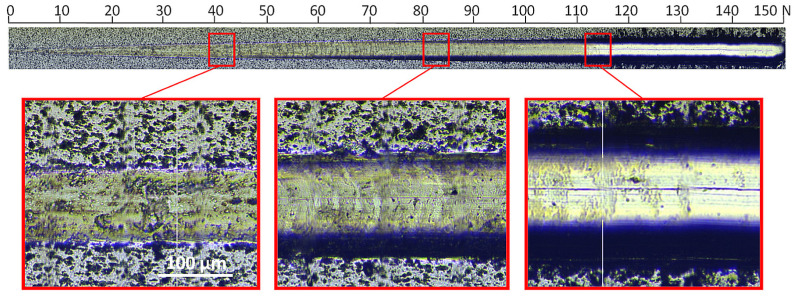
Optical micrograph of the scratch path of the V-O(0)-N coating, made under a load of up to 150 N and selected critical failure areas.

**Figure 9 materials-17-00419-f009:**
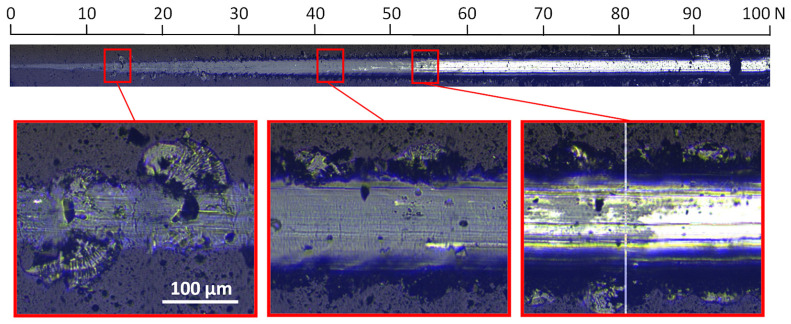
Optical micrograph of the scratch path of the V-O(20)-N coating, made under a load of up to 100 N and selected critical failure areas.

**Figure 10 materials-17-00419-f010:**
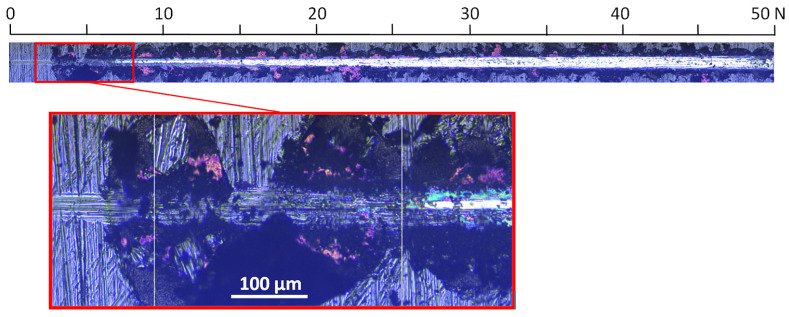
Optical micrograph of the scratch path of the V-O(50)-N coating, made under a load of up to 50 N and selected critical failure area.

**Figure 11 materials-17-00419-f011:**
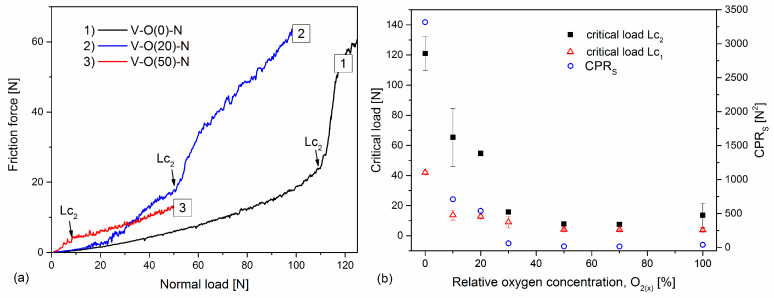
The friction force is dependent on normal load (**a**) and critical loads (Lc_1_ and Lc_2_) and CPR_s_ parameter of V-O-N coatings (**b**) formed at various relative oxygen concentrations.

**Figure 12 materials-17-00419-f012:**
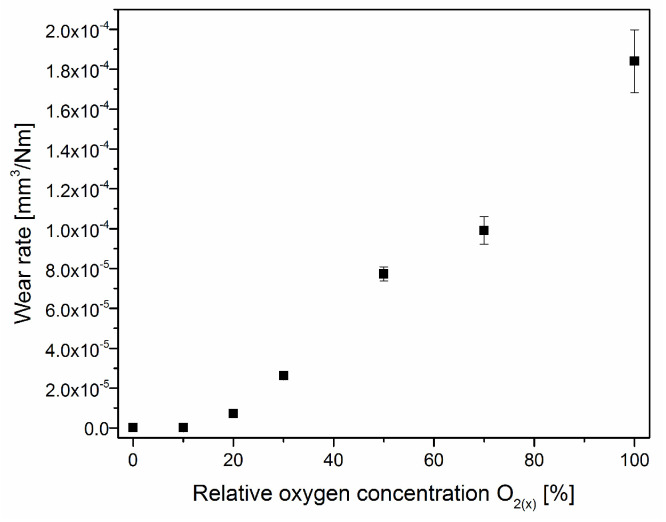
The wear rate of V-O-N coatings formed at various relative oxygen concentrations.

## Data Availability

All data are provided in the article.

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
