# Peer review of "Mechanical Properties of V-O-N Coatings Synthesized by Cathodic Arc Evaporation"

_materials, 2024, doi:10.3390/ma17020419_

Round 1

Reviewer 1 Report

Comments and Suggestions for Authors

Interesting investigations, but a phase diagram of the V-O-N three-substance system would be easier to understand.

There is a lack of concrete data on the investigation with electron microscopy and EDS, detector, high voltage, focus distance .... 

There is a lack of concrete data on the investigation with X-ray diffraction, such as high voltage, apertures, diffractometer arrangement, step size and counting time. 

Figures 2, 4, 7, 11, 12, 13 should be divided into two parts a and b if possible to save space.  

Figure 5 Diffractograms are shown here, not spectra! For better readability, scale the figure to the full width of the page. 

Comments on the Quality of English Language

Quality English is good, find no obvious errors

Author Response

Dear Reviewer,

thank you for your great work in reading and commenting on the manuscript. Added items are marked in red, deleted items are crossed out and highlighted in yellow.

  1. A phase diagram of the V-O-N three-substance system would be easier to understand.

Answer: As per comment 5, we have enlarged Figure 5 for better readability. In our opinion, three different symbols: an inverted triangle, a circle and a rhombus characterized by individual colors: magenta, red and blue, respectively, clearly characterize the three emerging phases - hexagonal VN (Figure 5a), cubic VN0.52O0.26 (Figure 5b.c) and V5O9 (Figure 5d-g).

  1. There is a lack of concrete data on the investigation with electron microscopy and EDS, detector, high voltage, focus distance .... 

Answer: It is added to the manuscript, please see the text. The microstructure and chemical composition of the coatings were investigated by scanning electron microscope (JSM-7001F) equipped with EDS (Energy Dispersive X-ray Spectroscopy) OXFORD Instruments INCA ENERGY 350 (20 kV).

  1. There is a lack of concrete data on the investigation with X-ray diffraction, such as high voltage, apertures, diffractometer arrangement, step size and counting time. 

Answer: (DRON-4-07, U = 35 kV, I = 20 mA) with a Cu-Kα radiation λ = 0.154187 nm and a nickel selectively absorbing filter, focusing by Bragg-Brentano, step size 0.05°, counting time 10 s.

  1. Figures 2, 4, 7, 11, 12, 13 should be divided into two parts a and b if possible to save space.  

Answer: We have a big problem with this comment. In accordance with the Journal's recommendations in Instructions for Authors, "Figures (and Tables) should be placed in the main text near to the first time they are cited." This makes it rather difficult to combine the mentioned Figures. Only two of them, Figures  11 and 12 can be connected, as can be seen in the text.

  1. Figure 5 Diffractograms are shown here, not spectra! For better readability, scale the figure to the full width of the page. 

Answer: Thank you for this comment. We have corrected the word "spectra" to "diffractograms". We have also significantly enlarged Figure 5 to make it more readable.

Reviewer 2 Report

Comments and Suggestions for Authors

The deposition rate shown in Fig. 2, it did not show any scientific progress. It did not express any measurement procedure. The authors may consider to delete this sub-section or present more information related to measurement information. Either one may make the paper better,

Author Response

Dera Reviewer,

thank you for your great work in reading and commenting on the manuscript. Added items are marked in red, deleted items are crossed out and highlighted in yellow.

The deposition rate shown in Fig. 2, it did not show any scientific progress. It did not express any measurement procedure. The authors may consider to delete this sub-section or present more information related to measurement information. Either one may make the paper better.

Answer: Chapter 3 RESULTS presents the coating test results. Here in Fig. 2, the deposition rate calculated as a coating thickness measured by Calotest divided by deposition time. This may be presented the thickness of the coating, but the deposition rate is often described in relevant publications, for example:

Farkas, N.; Zhang, G.; Ramsier, R.D.; Evans, E.A.; Dagata, J.A. Characterization of Zirconium Nitride Films Sputter Deposited with an Extensive Range of Nitrogen Flow Rates. Vac. Sci. Technol. A 2008, 26, 297–301.

Laurikaitis, M.; Dudonis, J.; Milˇcius, D. Deposition of zirconium oxynitride films by reactive cathodic arc evaporation and investigation of physical properties. Thin Solid Films 2008, 516, 1549–1552.

Rizzo, A.; Signore, M.A.; Mirenghi, L.; Di Luccio, T. Synthesis and characterization of titanium and zirconium oxynitride coatings. Thin Solid Films 2009, 517, 5956–5964.

Da Silva Oliveira, C.I.; Martinez-Martinez, D.; Cunha, L.; Rodrigues, M.S.; Borges, J.; Lopes, C.; Alves, E.; Barradas, N.P.; Apreutesei, M. Zr-O-N coatings for decorative purposes: Study of the system stability by exploration of the deposition parameter space. Coat. Technol. 2018, 343, 30–37.

Akash Singh, P. Kuppusami, Shabana Khan, C. Sudha, R. Thirumurugesan, R. Ramaseshan, R. Divakar, E. Mohandas, S. Dash, Influence of nitrogen flow rate on microstructural and nanomechanical properties of Zr–N thin films prepared by pulsed DC magnetron sputtering, Applied Surface Science 280 (2013) 117– 123

Reviewer 3 Report

Comments and Suggestions for Authors

The paper deals on the formation of vanadium oxynitride via cathodic arc evaporation and the effect of variable amounts of oxygen on its composition and mechanical properties.

In my opinion the paper can be considered for publication once some imprecisions are removed. Below the list of my concerns.

Abstract (row 40th): "Due to the higher reactivity of oxygen compared to nitrogen, ionic metal-oxygen bonds are formed even at low nitrogen concentrations." probably it is "...low oxygen concentrations". 

Introduction chapter. Please better focus on the state of the art and the present research aim.

Figure 1. Undoubtedly the deposited layers appares different. The Authors attribute these differences to the different chemical content. May be it is correct. However, it is well known that some transition metal oxides (TiO2 , ZrO2 and so on) present strong interferential colours depending upon their thickness. Since the layers deposited at different oxygen content strongly differ in thickness the potential role of interferential colour can not be ruled out. Since aesthetic apparence is considered a valuable characteristic, I would suggest to evaluate coatings of the same thickness to assure it. 

Figure 4. Please add standard deviation of compositional data.

Author Response

Dear reviewer,

thank you for your great work in reading and commenting on the manuscript. Added items are marked in red, deleted items are crossed out and highlighted in yellow.

  1. Abstract (row 40th): "Due to the higher reactivity of oxygen compared to nitrogen, ionic metal-oxygen bonds are formed even at low nitrogen concentrations." probably it is "...low oxygen concentrations". 

Answer: Thank you for this comment. This is an obvious mistake, the sentence makes no sense. We have replaced "nitrogen" with "oxygen", please see in the text.

  1. Introduction chapter. Please better focus on the state of the art and the present research aim.

Answer: Thank you for this comment. We added some sentences to Introduction, please see below:

" However, the amount of data on the mechanical properties of V-O-N coatings is small and there is no information on the hardness, adhesion of the coatings and their wear resistance, parameters taken into account when applying the coatings as decorative ones. It is known that doping the VN coating with C [22] or Ag [23] causes a decrease in hardness, but the tribological properties are completely opposite.This causes difficulties in predicting the mechanical properties of oxygen-doped VN coatings. The change in the structure of V-N coatings as a result of their oxidation is the most frequently studied [24]. V-O-N coatings have been formed so far by the magnetron sputtering method [21].

Due to their properties, thin vanadium nitride coatings are used as protective and lubricating coatings. Oxidation of vanadium nitride occurring in the friction process (especially at high temperatures) significantly affects the tribological properties [25,26]."

  1. Figure 1. Undoubtedly the deposited layers appares different. The Authors attribute these differences to the different chemical content. May be it is correct. However, it is well known that some transition metal oxides (TiO2 , ZrO2 and so on) present strong interferential colours depending upon their thickness. Since the layers deposited at different oxygen content strongly differ in thickness the potential role of interferential colour can not be ruled out. Since aesthetic apparence is considered a valuable characteristic, I would suggest to evaluate coatings of the same thickness to assure it. 

Answer: Indeed, there are authors pointing out that the color of coatings may change with their thickness (interference influence). However, this applies to coatings with small thicknesses, well below 1 µm. The coatings tested in this manuscript are characterized by a thickness of over 3.5 µm, so this effect is not important. Vaz's works [e.g. Vaz, F.; Carvalho, P.; Cunha, L.; Rebouta, L.; Moura, C.; Alves, E.; Ramos, A.R.; Cavaleiro, A.; Goudeau, P.; Riviere, J.P. Property change in ZrNxOy thin films: Effect of the oxygen fraction and bias voltage. Thin Solid Films 2004, 469–470, 11–17.] include the study of coatings from 1.3 µm to 3.2 µm thick and the authors analyze the change in the properties of ZrON coatings (including color), ignoring the interference effect. Similarly, Da Silva [Da Silva Oliveira, C.I.; Martinez-Martinez, D.; Cunha, L.; Rodrigues, M.S.; Borges, J.; Lopes, C.; Alves, E.; Barradas, N.P.; Apreutesei, M. Zr-O-N coatings for decorative purposes: Study of the system stability by exploration of the deposition parameter space. Surf. Coat. Technol. 2018, 343, 30–37.], when examining ZrON coatings with a thickness of 418 nm to 1390 nm (including color), also does not take into account interference effects when determining the color of the resulting coatings.

The formation of the coatings characterized by uniform thickness is quite a time-consuming task. The thickness of the coating is not only related to the time of its formation, but also to technological parameters, including: gas flow rates (partial pressures in the working atmosphere), discharge power (magnetron sputtering) or arc current (cathode arc evaporation), substrate temperature, substrate bias voltage. Although testing of coatings of the same thickness would be valuable in assessing color, but also adhesion, hardness and friction, the results of such tests are extremely rare. An additional parameter influencing the above values is the surface roughness, which cannot be "uniformed" technologically.

4. Figure 4. Please add standard deviation of compositional data.

Answer: The measurement accuracy is approximately 0.5 at.% for metallic element and about 1.0 at.% for non-metallic elements. For this reason, they are too small to be visible in the drawing. The following sentence has been added in section 2.2. Coating investigations

"The elements were analyzed with an accuracy of about 0.5 at. % (vanadium) and 1.0 at. % (nitrogen and oxygen),"

Round 2

Reviewer 1 Report

Comments and Suggestions for Authors

the changes are fine.